# The Benefits of Practicing Physical Activity in the Aquatic Environment on Health and Quality of Life

**DOI:** 10.3390/healthcare13091053

**Published:** 2025-05-03

**Authors:** Gabriela Tomescu, Valeria Bălan, Kamer-Ainur Aivaz, Mihaela Zahiu

**Affiliations:** 1Faculty of Physical Education and Sport, Ovidius University of Constanța, 900470 Constanța, Romania; 2Faculty of Physical Education and Sport, National University of Physical Education and Sport, 060057 Bucharest, Romania; mihaela.zahiu@unefs.ro; 3Faculty of Economic Sciences, Ovidius University of Constanta, 900001 Constanta, Romania; kamer.aivaz@365.univ-ovidius.ro

**Keywords:** quality of life, health, physical activity, aquagym, aquatic programs, benefits

## Abstract

**Background/Objectives**: Aquagym is a form of practicing aerobic gymnastics to stay fit, which is constantly progressing in line with the dynamics of market growth and the fitness industry, bringing benefits to physical and mental health. The purpose of the study was to identify the level of popularity of this sport among people aged 20–65 years and to highlight the benefits of practicing aquagym. **Methods**: This quantitative research study was designed to use a structured questionnaire to collect data from 150 participants. The sample was chosen to reflect various age groups and to ensure representativeness for the general population interested in physical activity. Participants were divided by gender into two age categories, under 45 and over 45, in order to examine differences in preferences and perceptions between these age segments. For the analysis of between-group differences, we used the Chi-Square and Cramer’s V tests, and for the profile of respondents, a Two-Step Cluster Analysis was applied. **Results**: Regarding knowledge of the aquagym concept, there is a notable discrepancy between females and males, with significantly more women (54%) than men (18%) stating that they are familiar with this concept. Among the perceived benefits of water gymnastics, most responses indicate maintaining muscle tone (25% women and 14% men) and a fun way to stay fit (20% women and 11% men). Young people under 45 years of age tend to attach more importance to the social and entertaining aspects of aquagym. As for the improvement in quality of life by practicing water gymnastics, a large number of respondents (82%) rated this contribution with a score of 4 or 5 on a 5-point scale, meaning that the positive impact of this activity was highly appreciated. **Conclusions**: Aquagym lessons are perceived as both an effective form of exercise for health improvement and a means of socialization. The outcomes of the study highlight the opportunity to continuously promote and develop gymnastics in the aquatic environment as an accessible and enjoyable form of physical activity for all ages.

## 1. Introduction

In recent years, the promotion of physical activity as a means to support health and improve quality of life has gained increasing attention among researchers and practitioners. Within this context, aquatic environments have emerged as valuable settings for physical exercise, offering unique physiological and psychological benefits. One of the most popular and accessible forms of aquatic exercise is aquagym, a type of aerobic gymnastics performed in water that combines the principles of fitness training with the therapeutic advantages of reduced joint impact and increased resistance.

Aquagym has evolved alongside the expansion of the global fitness industry, adapting to the diverse needs and expectations of modern society. Beyond its capacity to improve muscle tone, cardiovascular endurance, and overall physical fitness, aquagym also supports mental well-being, social engagement, and enjoyment—factors that contribute to the development of lifelong exercise habits.

Despite these well-documented benefits, public awareness and participation in aquagym programs remain uneven, often influenced by age, gender, and prior exposure to aquatic activities. Understanding how different demographic groups perceive and experience aquagym is essential for developing inclusive and effective health promotion strategies.

The present study aimed to explore the level of popularity of aquagym among individuals aged 20 to 65 years and to identify the perceived benefits of practicing this form of exercise. By analyzing the preferences and perceptions of participants from different age and gender groups, this study sought to provide data-driven insights that can guide the development and promotion of aquatic fitness programs adapted to a wide range of needs and expectations.

Water is a captivating environment that attracts people of different ages and offers them various forms of motor activities. Initially considered an alternative to land-based exercise, in-water movement has now become one of the best ways to practice aerobic exercise, as it educates fitness components without causing trauma, pain, and suffering, stimulates the body’s ability to adapt to exercise [1], and can be practiced in the form of aquatic sports and aquatic programs with a major impact on both health and well-being [2]. Aquatic sports, whether competitive or as leisure activities, involve competition, fun, and relaxation. In contrast, aquatic programs put an emphasis on participants’ well-being, with quality of life at the forefront [3]. From aquatic education programs for young children to aquatic programs adapted to overweight people, pregnant women or the elderly, all of them prioritize the individual and maintaining an optimal fitness level needed to perform daily activities in good physical condition [4].

The purpose of this study was to explore the perceived benefits of aquagym programs as experienced and reported by their participants. While the existing literature highlights the significant physical and mental health outcomes associated with aquatic exercise, our research focuses specifically on the personal perspectives of those who actively engage in aquagym. We aimed to assess the popularity of this form of physical activity within the general population and to better understand why individuals may choose water-based exercise over traditional land-based alternatives. More specifically, the study sought to determine the extent to which aquagym is known, the level of awareness regarding its benefits, the elements that contribute to its appeal, and the overall interest in further developing this type of physical activity.

## 2. Literature Review

### 2.1. The Advantages of Practicing Aquatic Gymnastics

Water gymnastics programs (aquagym, cardio-aquagym, aquabuilding, aquastretching, aqua-aerobics, aquatic hip-hop) combine swimming movements with gymnastics movements in attractive choreography performed to music [5]. Music is appropriate for each part of the lesson, being recommended in sports activities because it sets the work rhythm and tempo, creates a pleasant atmosphere and engages participants in the activity throughout the program [6]. The advantages of attending music and dance-based aerobic programs for different age categories are very important: music can cause positive emotions, reduce unpleasant sensations during physical exertion and fatigue, improve work efficiency [6], while dance programs contribute to the development of psychomotor and coordination skills [7], stimulate multiple intelligences [8], improve muscle tone with a direct impact on body posture [9,10], reduce lower back pain [11], and contribute to harmonious physical development [12]. Practicing physical exercises to a background music improves the emotional and mental state of practitioners [13]. Music, along with dancing, reduces depression, aggressive behavior, symptoms associated with dementia, and improves cognitive ability [14].

Aquatic gymnastics, also known as water aerobics or hydrogymnastics, has gained popularity as a form of exercise that combines the benefits of water-based activities with the structured movements of gymnastics. Aquatic gymnastics offers several physical benefits, particularly due to the unique properties of water, such as buoyancy and resistance [15]. These properties make it an ideal option for individuals with joint issues or those seeking low-impact exercise.

Aquatic gymnastics enhances functional fitness, which is the ability to perform daily activities with ease [16]. Studies have shown that aquatic exercises, such as water aerobics and deep-water running, improve functional outcomes like the 30 s chair stand, 30 s arm curl, and Timed Up-and-Go test [17]. These improvements are comparable to or even superior to those achieved through traditional gymnastics and swimming.

Aquatic gymnastics is an effective way to improve cardiovascular health. High-intensity interval training (HIIT) in water, such as aquatic HIIT, has been shown to enhance maximal aerobic capacity, maximal heart rate, and metabolic equivalents (METs) [18]. These benefits are similar to those achieved through swimming and traditional gymnastics but with the added advantage of reduced joint stress.

The resistance provided by water in aquatic gymnastics helps build muscle strength and endurance. For example, deep-water running and water aerobics have been shown to improve lower limb muscle endurance and reduce resting heart rate [17]. These effects are particularly beneficial for older adults and individuals with chronic conditions.

Aquatic gymnastics can aid in weight management by improving body composition. Studies have found that aquatic exercises reduce body mass, waist circumference, and fat mass while increasing lean body mass [19]. These changes are similar to those observed in traditional gymnastics and swimming but may be more sustainable for individuals who prefer low-impact activities.

In addition to physical benefits, aquatic gymnastics offers several mental health advantages, making it a holistic form of exercise.

Aquatic exercises have been shown to reduce symptoms of anxiety and depression. A systematic review and meta-analysis found that aquatic aerobics significantly improved mental health, with a standardized mean difference (SMD) of −0.92 for anxiety and depression symptoms [20]. These effects are comparable to those achieved through swimming and traditional gymnastics.

Aquatic gymnastics may also enhance cognitive function, particularly in older adults. Studies have found that aquatic exercises improve executive functions and memory, possibly due to the structural and functional changes in the fronto-hippocampal axis [21]. These cognitive benefits are similar to those observed in swimming and traditional gymnastics.

The immersive nature of water-based exercises contributes to improved mood and overall well-being. A study on older women found that water-based fitness programs enhanced subjective well-being, self-perceived mental health, and physical health status [22]. These effects are often more pronounced in aquatic settings compared to land-based activities.

The buoyancy and sensory experience of water provide a calming effect, making aquatic gymnastics an effective stress-relief activity. Studies have shown that aquatic exercises reduce stress hormones and improve sleep quality, which are essential for mental health [23,24].

Aquatic gymnastics offers a unique combination of physical and mental benefits that make it an excellent alternative to traditional gymnastics and swimming. Its low-impact nature, coupled with the therapeutic properties of water, makes it particularly suitable for older adults, individuals with chronic conditions, and those seeking a holistic exercise experience [25,26]. While traditional gymnastics and swimming also provide significant benefits, aquatic gymnastics stands out for its ability to improve functional fitness, reduce joint stress, and enhance mental well-being [20,21,22].

### 2.2. Types of Water Gymnastic Programs and Their Effects on Health

The forms of practicing water gymnastic programs have constantly progressed due to the development of the fitness industry and market demands [27], influenced by other forms of exercise in the aquatic environment performed for recreational, sporting and therapeutic purposes [5]. Thus, water gymnastics programs currently include numerous exercise variations such as aquafitness, aquagym, hydrobike, aquastretching, aquastep, and aquaboxing [27]. Incorporating aquagym exercises into rehabilitation programs for individuals with physical disabilities offers numerous benefits, enhancing both physical and psychological well-being [28]. The buoyancy of water reduces strain on joints, allowing for safer and more effective exercise. This approach not only improves physical fitness but also fosters social interaction and emotional health. Stan (2013) proposes an aquatic intervention plan for people with neuromuscular disorders [28]. The author clearly describes the structure of the program, its stages and therapeutic goals, providing an applied perspective on how aquatic activities can be adapted according to the needs of the participants. In another study, Stan (2012) discusses aquatic fitness and rehabilitation among people with disabilities, highlighting the physical and psychological benefits of these interventions and emphasizing the importance of exercise customization and continuous progress monitoring [29]. Another relevant example is provided by Henriques, Henriques and Mirco (2013), who explore the use of aquatic exercise in the rehabilitation of children and adolescents with cerebral palsy [30]. The research highlights both the structure of the sessions and how they contribute to improved motor function. Similarly, Escobar et al. (2013) present a case study examining the impact of an aquatic fitness program on the mobility and general health of a woman with intellectual and physical disabilities, using functional indicators to assess progress [31].

The authors Badău and Badău compared aqua ludotherapy, aquagym, and therapeutic swimming, using a questionnaire to identify the benefits obtained from their practice. The conclusions indicate that aquagym improves motor capacity, aqua ludotherapy has an important action on a psychological level, and therapeutic swimming contributes to recovery and restoration [12]. Other researchers demonstrated aquagym improved joint mobility and muscle strength [32], as well as increased aerobic capacity, neuromuscular coordination and stress resistance [33]. Laboratory studies have shown that aquarunning can replace heavy running workouts practiced on land, as movements are made easier in water [34]. Elderly people who practiced aquajogging had improvements in health and fitness components [35], feeling less pain in their joints [36]. Also, significant effects have been recorded after the constant practice of aquastretching and aquapilates programs used in medical rehabilitation programs, such as strengthening of muscles and increased range of motion in joints [37].

More recently, Lobanov and colleagues (2022) investigated the effects of aquatic exercise on the walking function in patients with post-COVID-19 sequelae. The study uses biomechanical tools to analyze gait stereotype improvement, providing insight into how recovery can be supported by aquatic interventions [38]. Last but not least, a paper by Faccini, Zanolli, and Dalla Vedova (2001) provides an extensive overview of aquatic therapy in the context of rehabilitation, explaining the components of the programs and the clinical evaluation methods used to measure effectiveness [39].

These examples demonstrate the diversity of aquatic interventions and the relevance of integrating rigorous evaluation tools, contributing to the scientific validation of the results obtained.

### 2.3. The Influence of Aquatic Programs on Physical and Physiological Aspects

It is known that thermal and chemical characteristics of water reduce depression [40], but the effects induced by the physical properties of water are found at physiological and psychological levels through specific positive cardiovascular, renal, respiratory, and muscular responses, but they also extend to metabolism and the central nervous system [41]. In water, body weight is apparently reduced, given that the Archimedes force acts upwards. The more the body is submerged in water, the greater the Archimedes force and the more obvious the reduction in body weight [42]. Aquagym improves the body composition [43,44], as it represents an effective motor activity to combat obesity [45,46]. All muscle groups are involved during an aquagym lesson, where even four to five thousand calories can be “burned” [47] in a shorter period of time compared to other physical activities [48]. This aspect was also demonstrated by Stan [49] in a study on children with obesity and physical problems, who participated in a specially designed water aerobics program. After applying the intervention program, it was found to be effective, as the children’s body composition changed, with visible effects on the other components of physical fitness.

Research in the field has demonstrated that water gymnastics is recommended for posttraumatic recovery [50], even for pregnant women [51] or those in the postnatal period. A 5-month study conducted by Navas et al. [52] showed that mothers improved their physical appearance and quality of life, with postnatal symptoms being reduced.

For an aquagym lesson to be efficient, it should be practiced 2 or 3 times a week for 45 to 50 min per session, and at least 20 complete exercises should be performed, according to the characteristics of the group concerned or the participant’s specific age, when working individually [53].

### 2.4. The Current Study

The literature emphasizes the impressive results of aquagym on the physical and mental health of its practitioners. In our research, we aimed to point out the benefits identified by aquagym participants themselves after attending this type of aquatic program. Therefore, the purpose of our study was to establish the level of popularity of this form of movement among the population and the effects of aquagym programs as recognized by participants who prefer water gymnastics over other forms of land-based exercise. Thus, the importance of the research was to determine whether the concept of aquagym is known, whether the benefits of practicing it are known, which aspects can increase the attractiveness of this activity, and whether the population is interested in developing this sport.

## 3. Materials and Methods

### 3.1. Research Design

This study employed a quantitative research design based on a structured questionnaire consisting of nine questions. Data were collected from a sample of 150 participants, carefully selected to represent a diverse range of age and gender groups, in order to ensure a representative cross-section of the general population interested in physical activity. The research was conducted over a three-month period, from 10 January to 15 March 2025.

The questionnaire was administered both online and in printed format to increase accessibility and participation. Respondents were informed about the voluntary and anonymous nature of the survey, and all responses were collected in accordance with ethical research standards. The questionnaire included both closed and semi-open questions, allowing for both quantitative analysis and the collection of short descriptive feedback.

The survey instrument was carefully developed based on the specific objectives of our study, which were grounded in existing literature emphasizing the significant physical and mental health benefits of aquagym. The questions were specifically designed to assess participants’ awareness and understanding of aqua aerobics, to explore the perceived benefits among those who have practiced it, to identify motivations for choosing aquatic programs, and to evaluate the perceived contribution of such activities to overall well-being. Each item was purposefully constructed to gather relevant and nuanced information from respondents in line with our research goals.

In the literature reviewed for this study, we did not identify any standardized or previously validated instruments that matched the specific objectives and structure of our research. As such, the questionnaire used in this study was developed by the authors specifically for this research context, based on themes and variables frequently discussed in the field of physical activity and aquatic exercise. We acknowledge that using a validated and reliable scale could have added further methodological rigor. However, given the exploratory nature of our study and the lack of existing standardized instruments adapted to the context of aquagym, we opted to develop custom questions that reflect the specific objectives of our research. Although the instrument has not been previously used in other studies under an established name, its content was carefully constructed to ensure internal consistency and clarity. The questionnaire items were inspired by commonly explored constructs in similar studies, and Likert-type scales were employed to support the quantitative analysis of attitudes and preferences. However, we consider it important to include or adapt validated instruments, where possible, in later phases of our research.

The study was grounded in the hypothesis that aquatic gymnastics programs would attract a significantly greater number of participants if the benefits of such activities were more widely recognized and promoted within society. Accordingly, the survey was designed to assess the level of popularity of aquagym, the public’s awareness of its benefits, and participants’ interest in engaging in this form of physical activity.

### 3.2. Participants

The questionnaire was administered in Romania to individuals aged between 20 and 65 years, residing in urban areas of the cities of Constanța and Bucharest. The study sample consisted of 150 participants, selected to reflect a balanced representation in terms of age (111 individuals under 45 years old and 39 over the age of 45) and gender (97 women and 53 men).

To explore potential differences in lifestyle, preferences, and perceptions regarding aquatic gymnastics programs, the participants were divided into two age groups: under 45 and over 45. This division facilitated a comparative analysis between younger and older respondents in terms of their awareness, interest, and engagement with water-based physical activities. After the age of 45, there is a transition into late adulthood and a person’s life can change considerably. There are other needs, preferences, and physiological and psychological changes, according to which each person chooses how to spend their leisure hours and what their goals are.

In addition to age, gender was also considered an important variable. Therefore, the responses were further analyzed to identify possible gender-based patterns or distinctions in attitudes toward aquagym. This approach facilitated a more comprehensive understanding of how demographic factors may influence the popularity and perceived benefits of aquatic exercise among urban populations in Romania. To strengthen the depth and generalizability of the study, additional demographic information, such as education level, occupation, and socio-economic background, could provide further insights and we will consider including these variables in future research.

Although we did not implement an aquagym program with our participants, our goal was to understand how widely aquatic activities are practiced, the motivations behind their practice, and whether such activities might appeal to individuals who have not yet explored this form of exercise. Additionally, we aimed to identify the specific characteristics of aquagym that could attract both people who have participated in aquatic gymnastics programs and individuals accustomed to other types of physical activity. The questionnaire results revealed that aquagym is less popular compared to other sports; however, respondents are aware of its benefits and perceive it as an appealing option. Many expressed interests in participating in aquagym programs, and believe that such activities can enhance quality of life and deserve greater promotion in society.

### 3.3. Data Collection

The primary instrument used for data collection was a self-administered questionnaire, specifically designed to capture relevant information about participants’ physical activity preferences and their experiences with aquagym. The questionnaire comprised nine items that addressed several key aspects: the types of physical activities most commonly practiced, the participants’ familiarity with and prior engagement in aquagym programs, the reasons for choosing this form of exercise, and their personal perceptions of its physical and psychological benefits.

To facilitate a meaningful analysis and interpretation, most items were constructed using a Likert-type response scale, allowing participants to express the degree of agreement or frequency related to each statement. This approach enabled the researchers to gain deeper insight into individual attitudes, motivations, and behaviors associated with aquatic physical activity.

The questionnaire was made available in both printed and digital formats, and it was distributed in public fitness centers, wellness clubs, and via online platforms targeting residents of Constanța and Bucharest. Participation was voluntary and anonymous, and respondents were informed about the confidentiality of their answers. This dual-mode distribution aimed to ensure broader accessibility and increase the reliability and diversity of the responses collected.

### 3.4. Instruments

The data analysis in this study was conducted using a set of statistical instruments aimed at evaluating the significance and strength of observed differences between age groups with respect to their preferences and experiences related to physical activity and aquagym. The Chi-Square test was employed to determine whether statistically significant differences existed in the distribution of responses across different age categories. To further assess the strength of these associations, Cramer’s V coefficient was calculated, offering a standardized measure of effect size that quantifies the degree of association between categorical variables—such as age group and preference for aquagym or other types of physical activity.

This rigorous statistical approach enabled the identification of both general patterns and specific differences among subgroups, offering valuable insights into how distinct age categories engage with physical activity. Such an analysis contributes to the development of evidence-based recommendations tailored to the unique health and fitness needs of various demographic groups.

Additionally, to deepen our understanding of participant profiles, we employed the Two-Step Cluster Analysis method. This multivariate technique allowed for the automatic grouping of respondents based on shared characteristics, facilitating a more nuanced examination of their behavior, motivations, and preferences. The method involves a two-phase process: in the first phase, a pre-clustering procedure groups similar cases using a scalable algorithm; in the second phase, these pre-clusters are refined and combined into final clusters using a distance-based criterion—such as log-likelihood or Euclidean distance—depending on the measurement level of the variables included.

A key advantage of the Two-Step Cluster method lies in its ability to automatically determine the optimal number of clusters based on the natural structure of the data, without requiring the researcher to predetermine this value. This feature minimizes subjectivity in the analysis and supports objective, data-driven segmentation of respondents, ensuring the reliability and validity of the findings.

## 4. Results

The responses received for the items proposed in the questionnaire helped us to determine to what extent the research participants considered that practicing water gymnastics could improve quality of life and to discover their opinions on the importance of promoting it in society. The questionnaire items and participants’ options, differentiated by gender and age, are detailed in Table 1 and Table 2.

### 4.1. Statistical Analysis According to Gender

The results shown in Table 1 provide a detailed overview of preferences and perceptions regarding the practice of water gymnastics (aquagym) but also other forms of leisure-time physical activity, according to gender.

The first dataset highlights a statistically significant difference between women and men as regards the preferred types of leisure-time exercise. The results show a strong preference of males for sports games (football, volleyball, handball, basketball, etc.), as the ratio is 23 men to 12 women. This suggests that males are more attracted to team sports and the related competition, while females tend to prefer less competitive activities such as dance and expression sports, given that 29 women chose this option compared to only 1 man. Interestingly, in terms of fitness, the distribution is almost equal, with 14 women and 13 men indicating similar acceptance of this type of exercise by both genders. The high value of Cramer’s V (0.505) shows a strong association between gender and preferences for certain types of movement.

Regarding knowledge of the aquagym concept, there is a notable discrepancy between females and males, with significantly more women (81) than men (27) stating that they are familiar with the concept. This difference indicates that aquagym is more popular and better promoted among women. However, when it comes to actually attending an aquagym lesson, the difference between those who participated and those who did not participate is smaller and statistically insignificant, with a *p*-value of 0.052 and a Cramer’s V of 0.159, suggesting that although the concept is known, participation is not very different between genders.

Responses regarding the purpose of practicing aquagym reflect various preferences, with a greater emphasis on muscle toning and harmonious physical development (22 women and 6 men), but also on fun and socialization (12 women and 5 men). These data point out that aquagym lessons are seen as both an effective form of exercise and an opportunity for socializing.

With respect to the perceived benefits of water gymnastics, most responses indicate maintaining muscle tone (38 women and 21 men) and a fun way to stay fit (30 women and 17 men). In contrast, improving interpersonal relationships and emotional well-being received a much smaller number of responses, indicating that these benefits are less recognized.

As for the improvement in quality of life by practicing water gymnastics, a large number of respondents (86) rated this contribution with 4 or 5 on a 5-point scale, meaning that the positive impact of this activity was highly appreciated.

Finally, the attraction to aquagym seems to be influenced by the pleasant aquatic environment (31 females and 18 males) and by the fact that the activity involves performing simple exercises with a low risk of injury and working on all muscle groups, these characteristics being appreciated by participants.

Responses to the question about constant participation in aquagym lessons and the importance of developing this concept suggest the existence of moderate but positive receptivity, emphasizing the potential of aquagym to become a more widespread and appreciated form of exercise in society.

### 4.2. Statistical Analysis by Age

The results shown in Table 2 provide an insight into preferences and behaviors related to practicing different forms of motor activity, segmented by age group (over and under 45 years). They allow us to understand how interests in certain types of exercise vary with aging.

Differences between age groups in terms of preferences for motor activities are not very pronounced, but there are indications that young people under 45 prefer more dynamic and competitive activities such as sports games (football, volleyball, handball, basketball, etc.), with 32 people in this age group playing them compared to only 3 in the older age group. This suggests that young people are more attracted to team sports, which involve a higher level of energy and interaction. Fitness and dance are significantly preferred by younger people, who have a propensity towards activities that involve both physical exercise and an expressive and socializing component.

Regarding interest in aquagym, which is known to both age groups, there are no significant differences in participation. About half of respondents, regardless of age, are aware of the concept, but very few have actually attended such lessons, with almost equal participation levels for the two groups.

The purposes for which participants choose to practice aquagym range from fun and socialization to physical recovery and muscle toning. However, differences between age groups are not statistically significant, which indicates that both groups find value in this sport for similar reasons.

Perceptions of the benefits of water gymnastics mainly focus on maintaining muscle tone and its ability to provide a fun and stress-free form of exercise. These two benefits are recognized and appreciated by both groups, but young people under 45 years of age tend to attach more importance to the social and entertaining aspects of aquagym.

As for the improvement in quality of life by practicing water gymnastics, both age groups agree on its positive impact, with young people under the age of 45 appreciating it the most, as reflected in their more enthusiastic responses.

Thus, although there is some variation between the preferences of age categories for different forms of exercise, both groups recognize and appreciate the benefits of water gymnastics for their health and quality of life. The outcomes of the study highlight the opportunity to continuously promote and develop aquagym as an accessible and enjoyable form of physical activity for all ages.

### 4.3. Two-Step Cluster Analysis Method

A thorough analysis of the questionnaire responses allowed us to create a certain profile of the research participants using the Two-Step Cluster Analysis method. We can thus conclude that although the aquagym concept is known to most of the people involved in our study, especially women under 45 years of age, many of them have never participated in an aquagym lesson. The questionnaire responses give us the opportunity to establish the profiles of three categories of people, as follows:

The first category is represented by women under 45 years of age, who know the aquagym concept and have participated in aquagym training for toning and harmonious physical development, but they like to practice fitness in their free time (Figure 1). Among the benefits of this activity, the fact that it is a fun way to stay fit makes these participants want to attend aquagym lessons on a regular basis, considering it important to develop this activity to improve quality of life. The attractiveness of this form of movement is ensured by the aquatic environment.

The second profile (Figure 2) represents the majority of women under the age of 45, who know the aquagym concept but have never participated in such a lesson. Since dance and other expression sports are at the top of preferences regarding the form of movement practiced in their free time, participants with this profile would like to attend aquagym lessons on a regular basis, considering it important to develop this activity and promote it in society, as they believe it could contribute to improving quality of life. Maintaining muscle tone is the benefit of water gymnastics that most respondents are interested in, and the attractiveness of this form of movement is ensured by the aquatic environment.

The last category of respondents describes the profile of males under 45 years of age, who have never practiced aquagym and are not familiar with this concept, preferring to play sports games in their free time. As shown in Figure 3, most of them do not consider that water gymnastics could greatly contribute to improving quality of life, but they believe it is important to promote this activity in society and would like to participate in aquagym lessons, especially because it is practiced in the aquatic environment and is a fun way to stay fit.

## 5. Discussion

The primary objective of our research was to investigate the level of public awareness and perception of aquagym, to identify the perceived benefits of practicing this form of exercise, and to explore the factors that may enhance its attractiveness and encourage wider participation. Given the well-documented physical and mental health benefits associated with aquagym, our broader aim was to highlight its potential as a valuable component of public health promotion.

Our study expresses the benefits identified by both aquagym participants and people practicing other leisure-time motor activities, who know, however, the benefits of this form of gymnastics. The results of our research indicate a low level of popularity of aquagym in society, but also a large number of people who would like this type of sport to be better highlighted and promoted. The benefits of practicing aquagym are recognized by people of both genders and of different ages, whether or not they practice this form of aquatic exercise. The present research confirms the results of previous studies that support the advantages of participating in aquagym lessons.

### 5.1. The Health Benefits of Practicing Water Gymnastics

All forms of aerobic gymnastics provide health benefits when practiced correctly, but those performed in water have some characteristics that cannot be found in a land-based environment. These features are due to body position in the water, exercise specificity, aquatic breathing, permanent water pressure on the body, and the apparent reduction in body weight, conditions that are only found in the aquatic environment [54].

Maintaining or improving physical fitness through a fun form of exercise increases the attractiveness to practicing it, 31% of our subjects being interested in this benefit. Water resistance to any movement, floating aids that increase the interest in exercises but are more demanding, as well as the pleasant environment created by water, light, and music, are favorable conditions for achieving improvements [55]. These aspects confirm the findings of other researchers [56], who also observed improvements in various components of physical fitness in active adult and adolescent women after participating in water gymnastics programs. Participation in aquagym lessons increases the strength of all muscles [55,57], especially the lower limbs [15]. This increase in strength is due to water resistance [6], which acts on all directions of movement. Obviously, strength training also has positive effects on improving physical condition [16,43].

The development of motor skills through recreational activities in the aquatic environment can contribute to improving human performance [58]. Water gymnastics programs also respond to the needs of people who do not feel comfortable performing land-based exercises [44] or are unable to perform them. The possibility to move more freely through the water was appreciated by a group of adolescent girls, whose interest in aquatic activities and the constant practice of aquagym increased [56].

Using water as an environment for recovery and prevention, older people can exercise to reduce stress on joints. Aquagym can improve the physical fitness of older adults, especially in terms of flexibility [59] and mobility of the upper and lower limbs—the Body Mass Index improved in 65% of subjects who participated in an aquagym program conducted by Marques Elias et al. [43]. This aspect is also known by our study subjects, who consider that maintaining muscle tone represents the greatest benefit of aquagym (39%). It also enhances coordination, agility, and balance [3], as the body position in the water can be vertical or horizontal [47], which gives these people independence in current activities [3].

### 5.2. The Structure and Effects of an Aquagym Lesson

To make the effects of aquagym programs visible, the lesson structure should follow the guidelines indicated by specialists. A warm-up, which lasts approximately 8 to 10 min [48], can be preceded by a series of land-based exercises aimed at preparing the body for the effort to come. The fundamental part of the aquagym lesson includes rhythmic, dynamic, and repetitive means that address as many muscle groups as possible and require aerobic systems to produce the energy needed for muscle contraction [60]. In the first part, moving through water by walking and jogging, as well as jumping, are recommended because they considerably improve blood oxygen supply, heart rate, and muscle strength [61]. These are followed by means aimed at developing the back and abdominal muscles [62]. Aquagym is a non-impact [12] or low-impact activity, and water supports body weight and facilitates movement [56,63,64]. The final part of the lesson, which lasts 5 to 8 min [48], allows for rapid recovery after exercise, with direct effects on body’s functional parameters that return to resting values. In this last part, a very important role is played by stretching and relaxation exercises, as they induce well-being [56,65].

Another point of view related to the structure of an aquagym lesson was expressed by Martinez-Moreno and Garcia-Pallares [9], who proposed a 30–45 min lesson, depending on participants’ characteristics, which would be divided into four parts: warm-up (8–10 min), aerobic sequence (15–20 min), exercises for muscle strength improvement (10–15 min), and relaxation sequence (5–8 min).

The results of our study reveal that aquagym lessons are seen as both an effective form of exercise and an opportunity for socializing, and are more popular and better promoted among women (54%) than men (18%). As for the improvement in quality of life by practicing water gymnastics, the research participants agree on its positive impact (82%), with young people under the age of 45 appreciating it the most. Although there is some variation between the preferences of age categories for different forms of exercise, both groups recognize and appreciate the benefits of aquagym for their health and quality of life. These findings highlight the opportunity to continuously promote and develop aquagym as an accessible and enjoyable form of physical activity for all ages. The fact that many people under 45 years of age tend to attach more importance to the social and entertaining aspects of aquagym represents an opportunity to spread this concept among youth through programs for men too, which are adapted to their needs and take into account that they prefer more dynamic and competitive sports, as suggested by their responses to our questionnaire. For this reason, we believe it is necessary to actively and consistently promote aquagym and other aquatic programs using innovative means adjusted to the trends of modern society.

The survey was designed to capture the general population’s familiarity with the concept of aquagym, motivations for engaging in aquatic activities, and the perceived impact of such programs on quality of life. The findings indicate that while aquagym remains less popular than other physical activities, there is a strong recognition of its benefits and a notable interest in participation.

## 6. Conclusions

The findings of this study confirm that aquagym is an accessible, enjoyable, and effective form of physical activity for individuals of all ages. It supports the development of lifelong exercise habits and helps overcome functional limitations, especially among older adults. Given its therapeutic and preventive benefits, aquagym should be more widely integrated into health promotion, rehabilitation, and fitness programs. We recommend the development of diverse aquatic exercise initiatives tailored to different age groups and needs.

### Study Limitations

While our findings suggest that aquagym, due to its innovative methodology, can play a significant role in the development of modern physical activity trends and research, there are several limitations that should be acknowledged. First, the limited number of existing studies on the benefits of aquatic programs—particularly for various social categories—contributes to the relatively low popularity of aquagym, especially among male participants.

Secondly, access to aquatic facilities remains significantly lower compared to opportunities for land-based sports. The lack of pools, combined with the overcrowding of existing facilities, restricts both public participation in aquagym and the ability to conduct large-scale research in this area. These infrastructural limitations affect the generalizability of our results, as the study sample may not be fully representative of the wider population.

Additionally, the study relied on self-reported data, which may be subject to response bias or inaccuracies in participant recall. Furthermore, the cross-sectional design does not allow for conclusions about causality or long-term effects of participation in aquagym programs. Future research should consider longitudinal designs and larger, more diverse samples to strengthen the validity and applicability of findings.

## Figures and Tables

**Figure 1 healthcare-13-01053-f001:**
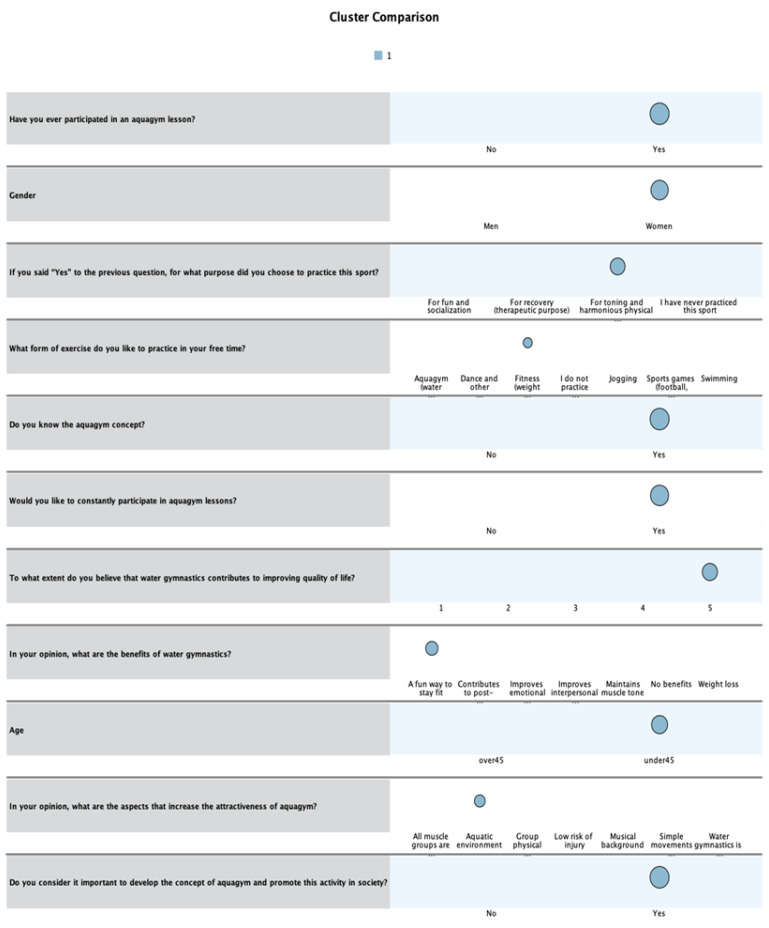
First profile of respondents. The color has no specific meaning in the Figure.

**Figure 2 healthcare-13-01053-f002:**
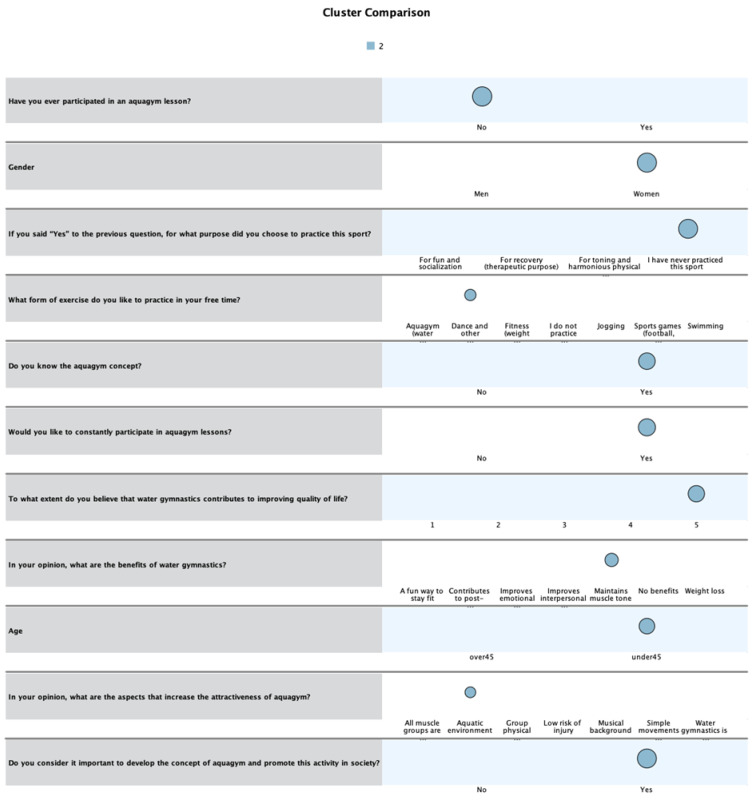
Second profile of respondents. The color has no specific meaning in the Figure.

**Figure 3 healthcare-13-01053-f003:**
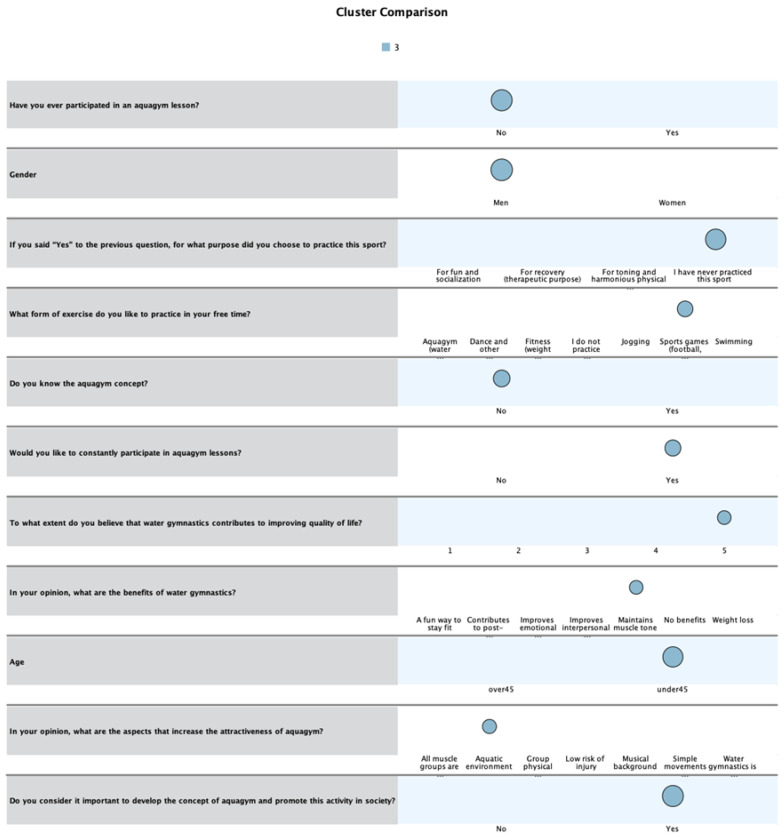
Third profile of respondents. The color has no specific meaning in the Figure.

**Table 1 healthcare-13-01053-t001:** Preferences and perceptions regarding the practice of water gymnastics (aquagym) but also other forms of leisure-time physical activity, according to gender.

		Women	Men	n	Chi-Square	*p*	Cramer’s V
What form of exercise do you like to practice in your free time?				150	38.208	<0.001	0.505
	Aquagym (water gymnastics to music)	8	0				
	Dance and other expression sports (gymnastics, skating, etc.)	29	1				
	Fitness (weight training)	14	13				
	Swimming	9	9				
	Sports games (football, volleyball, handball, basketball, etc.)	12	23				
	Jogging	9	3				
	I do not practice sports in my free time	16	4				
Do you know the aquagym concept?				150	18.025	<0.001	0.347
	Yes	81	27				
	No	16	26				
Have you ever participated in an aquagym lesson?				150	3.787	0.052	0.159
	Yes	35	11				
	No	62	42				
If you said “Yes” to the previous question, for what purpose did you choose to practice this sport?				150	3.658	0.301	0.156
	I have never practiced this sport	59	39				
	For fun and socialization	12	5				
	For recovery (therapeutic purpose)	4	3				
	For toning and harmonious physical development	22	6				
In your opinion, what are the benefits of water gymnastics?				150	5.717	0.456	0.195
	Contributes to post-traumatic recovery	16	8				
	Improves interpersonal relationships	1	0				
	Improves emotional well-being	6	1				
	Maintains muscle tone	38	21				
	No benefits	0	2				
	Weight loss	6	4				
	A fun way to stay fit	30	17				
To what extent do you believe that water gymnastics contributes to improving quality of life?				150	18.625	<0.001	0.352
	1	0	2				
	2	2	3				
	3	7	13				
	4	22	15				
	5	66	20				
In your opinion, what are the aspects that increase the attractiveness of aquagym?				150	19.764	0.003	0.363
	Group physical activity	13	6				
	Musical background	9	1				
	Aquatic environment	31	18				
	Water gymnastics is not attractive	0	3				
	Simple movements without muscle pain	18	1				
	Low risk of injury	8	9				
	All muscle groups are involved	18	15				
Would you like to constantly participate in aquagym lessons?				150	8.292	0.004	0.235
	Yes	78	31				
	No	19	22				
Do you consider it important to develop the concept of aquagym and promote this activity in society?				150	2.730	0.099	0.135
	Yes	94	48				
	No	3	5				

Note: n: number of participants; *p*: significance level.

**Table 2 healthcare-13-01053-t002:** Insight into preferences and behaviors related to practicing water gymnastics and different forms of motor activity, segmented by age group.

		Over 45	Under 45	n	Chi-Square	*p*	Cramer’s V
**What form of exercise do you like to practice in your free time?**				150	12.424	0.053	0.288
	Aquagym (water gymnastics to music)	3	5				
	Dance and other expression sports (gymnastics, skating, etc.)	9	21				
	Fitness (weight training)	5	22				
	Swimming	7	11				
	Sports games (football, volleyball, handball, basketball, etc.)	3	32				
	Jogging	3	9				
	I do not practice sports in my free time	9	11				
**Do you know the aquagym concept?**				150	0.634	0.426	0.065
	Yes	30	78				
	No	9	33				
**Have you ever participated in an aquagym lesson?**				150	0.000	0.987	0.001
	Yes	12	34				
	No	27	77				
**If you said “Yes” to the previous question, for what purpose did you choose to practice this sport?**				150	2.130	0.546	0.119
	I have never practiced this sport	24	74				
	For fun and socialization	3	14				
	For recovery (therapeutic purpose)	2	5				
	For toning and harmonious physical development	10	18				
**In your opinion, what are the benefits of water gymnastics?**				150	8.532	0.202	0.238
	Contributes to post-traumatic recovery	8	16				
	Improves interpersonal relationships	1	0				
	Improves emotional well-being	0	7				
	Maintains muscle tone	18	41				
	No benefits	0	2				
	Weight loss	3	7				
	A fun way to stay fit	9	38				
**To what extent do you believe that water gymnastics contributes to improving quality of life?**				150	9.086	0.059	0.246
	1	0	2				
	2	0	5				
	3	3	17				
	4	6	31				
	5	30	56				
**In your opinion, what are the aspects that increase the attractiveness of aquagym?**				150	6.715	0.348	0.212
	Group physical activity	3	16				
	Musical background	3	7				
	Aquatic environment	14	35				
	Water gymnastics is not attractive	0	3				
	Simple movements without muscle pain	8	11				
	Low risk of injury	2	15				
	All muscle groups are involved	9	24				
**Would you like to constantly participate in aquagym lessons?**				150	0.020	0.887	0.012
	Yes	28	81				
	No	11	30				
**Do you consider it important to develop the concept of aquagym and promote this activity in society?**				150	0.581	0.446	0.062
	Yes	36	106				
	No	3	5				

## Data Availability

The data presented in this study are available on request from the corresponding author.

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
