# Peer review of "The Benefits of Practicing Physical Activity in the Aquatic Environment on Health and Quality of Life"

_healthcare, 2025, doi:10.3390/healthcare13091053_

Round 1
Reviewer 1 Report
Comments and Suggestions for Authors
Abstract: Line 23. Results: The results are presented numerically. Comparative results should be given in percentages instead.
Keywords: Adequate.
Introduction: The section is too long (approximately two and a half pages). The introduction should be shortened.
Lines 153-158: Why are Table 1 and Table 2 included here?
What is the significance of the study? The introduction is very long, but the importance of the study could be better explained.
Participants: On what basis were the participants divided into two groups (under 45 years and over 45 years)?
The sample size appears sufficient.
Methodology: How were the survey questions determined? Instead, a validated and reliable scale could have been used.
Table 1: The capital letter "N" refers to the population, whereas the lowercase "n" should be used here.
Study Design: The participants in the study engage in different types of sports (Table 1). This contradicts the study's hypothesis. The group should consist solely of aquagym participants to be acceptable. Since these groups also participate in other sports, generalizing the results is not appropriate. This issue affects the study design.
Aquagym Participants: There is no information on how many days per week they engage in aquagym, the duration of each session, or how long they have been practicing it.
Lines 328-330: This information does not seem relevant to the study. The study focuses on quality of life, but definitions related to aerobic exercise or physical development are unnecessary.
Conclusions: Line 400. This section should be a single paragraph. It has been unnecessarily extended.
Comments on the Quality of English Language
Sufficient but minor revision.
Author Response
Thank you for the recommendations.
Please find the answers in the attachment.

Reviewer 2 Report
Comments and Suggestions for Authors
This study addresses the issue of physical activity in water with an attempt to explain its significance in the group of people studied. The introduction to this study is burdened with too much general knowledge on the subject.
The authors did not specify the area of choice of the people studied, what environment they came from, education, age in age groups. There is no information about participants in the chosen limit above and below 45 years of age.
The authors did not clearly specify the research goal and did not discuss it sufficiently, it was not clarified in the discussion either. Unfortunately, the research questionnaire contains logical errors.

Author Response

(The authors gave the same response as above.)

Reviewer 3 Report
Comments and Suggestions for Authors
Introduction
I recommend that authors thoroughly review their introduction. Please be to the point regarding the state of the art and background on the topic under review. Then, I recommend including research that has analyzed or implemented different aquatic programs, as well as describing their assessment instruments. Finally, conclude the introduction with a final paragraph describing the purpose of your study; do not include references to tables.
- Lines 43-44: Reference this statement
Water-based movement can be practiced in the form of aquatic sports and aquatic programs with a major impact on both health and well-being.
- Lines 47-49: Reference these statements
Aquatic sports, whether competitive or as leisure activities, involve competition, fun, and relaxation. In contrast, aquatic programs put an emphasis on participants' well-being, with quality of life at the forefront.
- Lines 53-55: Reference this statement
Water gymnastics programs, which fall into the category of aquatic programs, combine swimming movements with gymnastics movements in attractive choreography performed to music.
- Line 53: I recommend adding some examples of these aquatic programs.
Water gymnastics programs (…)
- Lines 68-70: Reference this statement
When practicing aquagym, the body position in the water can be vertical or horizontal, which requires a lot of balance, especially in situations where the starting positions of the exercises are different.
- Lines 120-121: I recommend adding some examples related to the advantages you mention. Then, reference those advantages.
“The advantages of attending music- and dance-based aerobic programs for different age categories are not to be neglected.”
Materials and Methods
Please add subsections to this section and complete them accordingly. As written, your study is eligible for rejection. Include sections such as: research design, participants, instrument (well scientifically justified), data collection and implementation procedure, and analysis.
Highlights:
- Provide valid and reliable information about the data collection instrument. Has it been used in any other study? What is its name? More scientific information is required.
Results
Order this section accordingly. Adding subsections would aid reading.
Consider placing the "Cluster Comparisons" in appendixes or rewording them in a more readable way. I recommend adjusting the size of the figures.
Discussion
I recommend that authors discuss this section in subsections, as it is difficult to establish a common thread between paragraphs.
Begin this section with the objective of your study.
- Lines 317-327: Discuss your data with that of other studies and give a constructive opinion about it.
- Lines 328-330 and 343-347: Same. I add these paragraphs as an example where it is recommended to discuss the authors.
Aquagym develops aerobic endurance, with some studies reporting significant improvements [31]. Other authors have identified minimal increases in this type of endurance [32], but their direct effects on participants’ daily lives were evident [33].
Aquagym can improve the physical fitness of older adults, especially in terms of flexibility [36] and mobility of the upper and lower limbs [32]. It also enhances coordination, agility and balance, which gives these people independence in current activities [2]. Archimedes’ force and hydrostatic pressure provide safety to aerobic gymnastics programs, which induces relaxation and positive functional and psychosocial effects [37].
- Lines 338-342: Please further link your statement so that it relates to the previous research you mention.
Gymnastics in the aquatic environment, due to its gentle nature, can be particularly effective in preventing frailty and cognitive decline associated with aging. Using water as an exercise environment reduces stress on joints, making it easier for people with various health conditions to perform movements and thus increasing accessibility to physical activity.
- Lines 358-376: Discuss this entire section accordingly. These contributions from the studies should have been summarized in the introduction…
- Line 392: Please reference these studies, if they are few, as indicated.
The small number of studies…
Conclusions
Authors are encouraged to be to the point with their conclusions.
Add some limitations of their study and practical applications.
Author Response

(The authors gave the same response as above.)

Round 2
Reviewer 2 Report
Comments and Suggestions for Authors
The reviewer notes many changes made to the manuscript that enhance its value.
Reviewer 3 Report
Comments and Suggestions for Authors